# PROXYLESSKD: DIRECT KNOWLEDGE DISTILLATION WITH INHERITED CLASSIFIER FOR FACE RECOGNITION

## ABSTRACT

Knowledge Distillation (KD) refers to transferring knowledge from a large model to a smaller one, which is widely used to enhance model performance in machine learning. It tries to align embedding spaces generated from the teacher and the student model (i.e. to make images corresponding to the same semantics share the same embedding across different models). In this work, we focus on its application in face recognition. We observe that existing knowledge distillation models optimize the proxy tasks that force the student to mimic the teacher's behavior, instead of directly optimizing the face recognition accuracy. Consequently, the obtained student models are not guaranteed to be optimal on the target task or able to benefit from advanced constraints, such as large margin constraint (e.g. margin-based softmax). We then propose a novel method named ProxylessKD that directly optimizes face recognition accuracy by inheriting the teacher's classifier as the student's classifier to guide the student to learn discriminative embeddings in the teacher's embedding space. The proposed ProxylessKD is very easy to implement and sufficiently generic to be extended to other tasks beyond face recognition. We conduct extensive experiments on standard face recognition benchmarks, and the results demonstrate that ProxylessKD achieves superior performance over existing knowledge distillation methods.

## 1 INTRODUCTION

Knowledge Distillation (KD) is a process of transferring knowledge from a large model to a smaller one. This technique is widely used to enhance model performance in many machine learning tasks such as image classification (Hinton et al., 2015), object detection (Chen et al., 2017b) and speech translation (Liu et al., 2019c). When applied to face recognition, the embeddings of a gallery are usually extracted by a larger teacher model while the embeddings of the query images are extracted by a smaller student model. The student model is encouraged to align its embedding space with that of the teacher, so as to improve its recognition capability.

Previous KD works promote the consistency in final predictions (Hinton et al., 2015), or in the activations of the hidden layer between student and teacher (Romero et al., 2014; Zagoruyko & Komodakis, 2016). Such an idea of only optimizing the consistency in predictions or activations brings limited performance boost since the student is often a small model with weaker capacity compared with the teacher. Later, Park et al. (2019); Peng et al. (2019) propose to exploit the correlation between instances to guide the student to mimic feature relationships of the teacher over a batch of input data, which achieves better performance. However, the above works all aim at guiding the student to mimic the behavior of the teacher, which is not suitable for practical face recognition. In reality, it is very important to directly align embedding spaces between student and teacher, which can enable models across different devices to share the same embedding space for feasible similarity comparison. To solve this, a simple and direct method is to directly minimize the L2 distance of embeddings extracted by student and teacher. However, this method (we call it L2KD) only considers minimizing the intra-class distance and ignores maximizing the inter-class distance, and is unable to benefit from some powerful loss functions with large margin (e.g. Cosface loss (Wang et al., 2018a), Arcface loss (Deng et al., 2019a)) constraint to further improve the performance.

In this work, we propose an effective knowledge distillation method named ProxylessKD. According to Ranjan et al. (2017), the classifier neurons in a recognition model can be viewed as the approximate embedding centers of each class. This can be used to guide the embedding learning as in this way, the classifier can encourage the embedding to align with the approximate embedding centers corresponding to the label of the image. Inspired by this, we propose to initialize the weight of the student's classifier with the weight of the teacher's classifier and fix it during the distillation process,

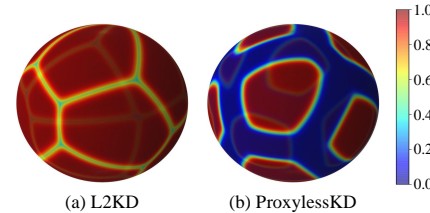

Figure 1: The embedding distributions extracted by (a) L2KD, and (b) ProxylessKD

which forces the student to produce an embedding space as consistent with that of the teacher as possible. Different from previous knowledge distillation works (Hinton et al., 2015; Zagoruyko & Komodakis, 2016; Romero et al., 2014; Park et al., 2019; Peng et al., 2019) and L2KD, the proposed ProxylessKD not only directly optimizes the target task but also considers minimizing the intra-class distance and maximizing the inter-class distance. Meanwhile it can benefit from large margin constraints (e.g. Cosface loss (Wang et al., 2018a) and Arcface loss (Deng et al., 2019a)). As shown in Figure 1, the intra-class distance in ProxylessKD combined with Arcface loss is much closer than L2KD, and the inter-class distance in ProxylessKD combined with Arcface loss is much larger than L2KD. Thus it can be expected that our ProxylessKD is able to improve the performance of face recognition, which will be experimentally validated.

The main contributions in this paper are summarized as follows:

- We analyze the shortcomings of existing knowledge distillation methods: they only optimize the proxy task rather than the target task; and they cannot conveniently integrate with advanced large margin constraints to further lift performance.

- We propose a simple yet effective KD method named ProxylessKD, which directly boosts embedding space alignment and can be easily combined with existing loss functions to achieve better performance.

- We conduct extensive experiments on standard face recognition benchmarks, and the results well demonstrate the effectiveness of the proposed ProxylessKD.

## 2 RELATED WORK

**Knowledge distillation.** Knowledge distillation aims to transfer the knowledge from the teacher model to a small model. The pioneer work is Buciluǎ et al. (2006), and Hinton et al. (2015) popularizes this idea by defining the concept of knowledge distillation (KD) as training the small model (the student) by exploiting the soft targets provided by a cumbersome model (the teacher). Unlike the one-hot label, the soft targets from the teacher contain rich related information among classes, which can guide the student to better learn the fine-grained distribution of data and thus lift performance. Lots of variants of model distillation strategies have been proposed and widely adopted in the fields like image classification (Chen et al., 2018), object detection (Chen et al., 2017a), semantic segmentation (Liu et al., 2019a; Park & Heo, 2020), etc. Concretely, Zagoruyko & Komodakis (2016) proposed a response-based KD model, Attention Transfer (AT), which aims to teach the student to activate the same region as the teacher model. Some relation-based distillation methods have also been developed, which encourage the student to mimic the relation of the output in different stages (Yim et al., 2017) and the samples in a batch (Park et al., 2019). The previous works mostly optimize the proxy tasks rather than the target task. In this work, we directly optimize face recognition accuracy by inheriting the teacher's classifier as the student's classifier to guide the student to learn discriminative embeddings in the teacher's embedding space. In (Deng et al., 2019b), they also directly copy and fix the weights of the margin inner-product layer of the teacher model to the student model to train the student model and the motivation of (Deng et al., 2019b) is the student model can be trained with better pre-defined inter-class information from the teacher model. However, different from (Deng et al., 2019b), we firstly analyze the shortcomings of existing knowledge distillation methods. Specifically, the existing methods target optimizing the proxy task rather than

the target task; and they cannot conveniently integrate with advanced large margin constraints to further lift performance. These valuable analyses and observations are not found in (Deng et al., 2019b) and other existing works. Secondly strong motivation and the physical explanation of the proposed ProxylessKD is well explained in our work. Figure 1 and corresponding analysis explained why ProxylessKD can achieve better performance than the existing methods that optimize the proxy task. Such in-depth analysis and strong physical explanation are novel and cannot be found in (Deng et al., 2019b) and other existing works. We believe these novel findings and the proposed solution are valuable to the face recognition community and will inspire researchers in related fields. Finally, solid experiments are designed and conducted to justify the importance of directly optimize the final task rather than the proxy task when doing knowledge distillation. And the properties of ProxylessKD about using different margin-based loss function and hyper-parameters are well examined. These detailed analyses about ProxylessKD cannot be found in (Deng et al., 2019b) and other existing works. We believe the above important differences and novel contributions make our work differs from (Deng et al., 2019b) and existing works.

**Loss functions used in face recognition.**    Softmax loss is defined as the pipeline combination of the last fully connected layer, softmax function, and cross-entropy loss. Although it can help the network separate categories in a high-dimensional space, for fine-grained classification problems like face recognition, it offers limited accuracy due to the considerable inter-class similarity. Liu et al. (2017) proposed Sphereface to achieve smaller maximal intra-class distance than minimal inter-class distance, which can directly enhance feature discrimination. Compared with SphereFace in which the margin $m$ is multiplied on the angle, Wang et al. (2018a); Whitelam et al. (2017) proposed CosFace, where the margin is directly subtracted from cosine, achieving better performance than SphereFace and relieving the need for joint supervision from the softmax loss. To further improve feature discrimination, Deng et al. (2018) proposed the ArcFace that utilizes the arc-cosine function to calculate the angle, i.e. adding an additive angular margin and back again by the cosine function. In this paper, we combine our ProxylessKD with the above loss functions to further lift performance, e.g. Arcface loss function.

# 3 METHODOLOGY

We first revisit popular loss functions in face recognition in Sec. 3.1, and elaborate on our ProxylessKD in Sec. 3.2. Then we introduce how to combine our method with existing loss functions in Sec. 3.3.

## 3.1 REVISIT LOSS FUNCTION IN FACE RECOGNITION

The most classical loss function in classification is the Softmax loss, which is represented as follows:

$$L_1 = -\frac{1}{N}\sum_{i=1}^{N} log \frac{e^{s \cdot cos(\theta_{w_y,x_i})}}{e^{s \cdot cos(\theta_{w_y,x_i})} + \sum_{k \neq y}^{K} e^{s \cdot cos(\theta_{w_k,x_i})}}. \tag{1}$$

Here, $w_k$ denotes the weight of the model classifier, where $k \in \{1, 2, ..., K\}$ and $K$ denotes the number of classes. $x_i$ is the embedding of $i$-$th$ sample and usually normalized with magnitude replaced with a scale parameter of $s$. $\theta_{w_k,x_i}$ denotes the angle between $w_k$ and $x_i$. $y$ is the ground truth label for the input embedding $x_i$. $N$ is the batch size. In recent years, several margin-based softmax loss functions (Liu et al., 2017; Wang et al., 2017; 2018a; Deng et al., 2019a) have been proposed to boost the embedding discrimination, which is represented as follows:

$$L_2 = -\frac{1}{N}\sum_{i=1}^{N} log \frac{e^{s \cdot f(m, \theta_{w_y,x_i})}}{e^{s \cdot f(m, \theta_{w_y,x_i})} + \sum_{k \neq y}^{K} e^{s \cdot cos(\theta_{w_k,x_i})}}. \tag{2}$$

In the above equation, $f(m, \theta_{w_y,x_i})$ is a margin function. Precisely, $f(m, \theta_{w_y,x_i}) = cos(m \cdot \theta_{w_y,x_i})$ is A-Softmax loss proposed in (Liu et al., 2017), where $m$ is an integer and greater than zero. $f(m, \theta_{w_y,x_i}) = cos(\theta_{w_y,x_i}) - m$ is the AM-Softmax loss proposed in Wang et al. (2018a) and the hyper-parameter $m$ is greater than zero. $f(m, \theta_{w_y,x_i}) = cos(\theta_{w_y,x_i} + m)$ with $m > 0$ is Arc-Softmax introduced in Deng et al. (2019a), which achieves better performance than the former. Fortunately, the proposed ProxylessKD can be combined with the above loss function, conveniently. In this paper, we combine our proposed ProxylessKD method with above loss functions and investigate their performance.

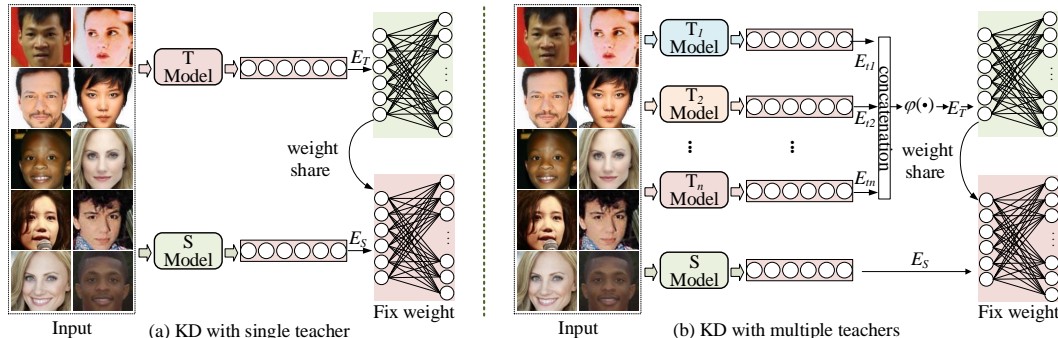

Figure 2: The diagram of proposed ProxylessKD. We firstly train a teacher model (a) or ensemble multiply trained teacher models (b), and then initialize the weight of the student's classifier with the weight of the teacher's or the ensembled teachers' classifier and fix the weight in the distillation stage.

### 3.2 INHERITED CLASSIFIER KNOWLEDGE DISTILLATION

The models trained for different devices are expected to share the same embedding space for similarity comparison. However, most existing knowledge distillation models only optimize proxy tasks, encouraging the student to mimic the teacher's behavior, instead of directly optimizing the target accuracy. In this paper, we propose to directly optimize the target task by inheriting the teacher's classifier to encourage better embedding space consistency between the student and the teacher.

**Knowledge distillation with single teacher.** In most of the existing distillation works (Chen et al., 2018; Liu et al., 2019a; Park & Heo, 2020), single large model is utilized as the teacher to guide the student. Hence, we firstly introduce our ProxylessKD method under the common single teacher model knowledge distillation, as shown in Figure 2 (a), where $E_T$ and $E_S$ represent the embedding extracted by the teacher model $T$ and the student model $S$, respectively. Unlike previous works (Hinton et al., 2015; Zagoruyko & Komodakis, 2016) that optimize the proxy task, we emphasize on optimizing the target task. To this end, we try to directly align the embedding space between the teacher model and the student model. Specifically, we firstly train a teacher model with the Arcface loss (Deng et al., 2019a), and then initialize the weight of the student's classifier with the weight of the teacher's classifier and fix the weight in the distillation stage. Using Arcface loss enables our method to benefit from large margin constraints in the distillation procedure. The distillation form of Figure 2 (a) can be defined as follows:

$$L_3 = -\frac{1}{N}\sum_{i=1}^{N} log \frac{e^{s \cdot f(m, \theta_{W_y^t, x_i})}}{e^{s \cdot f(m, \theta_{W_y^t, x_i})} + \sum_{k \neq y}^{K} e^{s \cdot cos(\theta_{W_k^t, x_i})}} \tag{3}$$

$f(m, \theta_{w_y^t, x_i})$ is a margin function, $x_i$ is the the embedding of $i\text{-}th$ sample in a batch, $w_y^t$ and $w_k^t$ are classifier's weights from the teacher model, y is the class of the $x_i$, $k \in \{1, 2, 3..., K\}$ and $K$ is the number of classes in the dataset. $\theta_{w^t, x_i}$ is the angle between $w^t$ and $x_i$. $m$ denotes the preset hyper-parameter. When we adjust the value of $m$, the interval among different intre-class samples will be changed.

**Knowledge distillation with multiple teachers:** Using an ensemble of teacher models would further boost the performance of knowledge distillation, according to previous work (Asif et al., 2019). Therefore, we here introduce how to implement ProxylessKD with the ensemble of teacher models in Figure 2, which better aligns with a practical face recognition system. To do this, we firstly train n different teacher models to acquire n embeddings for each input and concatenate the n embeddings to produce a high-dimensional embedding. Secondly, we employee a dimensionality reduction layer or the PCA (Principal Component Analysis) method (Wold et al., 1987) to reduce the high-dimensional embedding to adapt to student's embedding dimensional. Finally, we input the embedding after dimensionality reduction into a new classifier and retrain it. We will do the same operate as the knowledge distillation with single teacher, when we obtain the new classifier. The

ensemble of teachers' classifier can be optimized as

$$E_T = \varphi(concatenation(E_{t_1}, E_{t_2}, ..., E_{t_n}))$$

$$L_4 = -\frac{1}{N}\sum_{i=1}^{N} log \frac{e^{s \cdot f(m, \theta_{W_y^t, E_T})}}{e^{s \cdot f(m, \theta_{W_y^t, E_T})} + \sum_{k \neq y}^{K} e^{s \cdot cos(\theta_{W_k^t, E_T})}} \quad (4)$$

$E_{t_i}, (i = 1, 2, ..., n), n \geq 2$ and $n \in \mathbb{N}^+$, is the embedding of the $i$-th sample extracted by n teacher models, $E_T$ is the dimensionality reduction vector of the $i$-th sample, $concatenation$ is the operation of concatenating embeddings, $\varphi$ is dimensionality reduction function (i.e., PCA function or the dimensionality reduction layer function).

## 3.3 INCORPORATING WITH OTHER LOSS FUNCTIONS

In Sec. 3.1, we only introduce the classic loss functions (i.e., Equation (1) and (2) in face recognition. As long as the loss function uses the output of the classifier to calculate the loss (e.g., ArcNegFace (Liu et al., 2019b)) for optimizing the network, the proposed ProxylessKD method can combine with it. Therefore, more powerful loss functions can be incorporated into our ProxylessKD to further improve the performance. The unified form can be defined as follows:

$$L = -\frac{1}{N}\sum_{i=1}^{N} \mathscr{C}(w^t, x_i) \quad (5)$$

$x_i$ is the embedding of the $i$-th sample. $w^t$ is the weight of the classifier from the teacher model. $N$ is the number of samples in a batch. We use $\mathscr{C}(\cdot)$ to represent the loss calculated by various loss function types. Note, it is not restricted to the field of face recognition but also applicable to other general classification tasks where our ProxylessKD is used for model performance improvement. For example, the recent work (Sun et al., 2020) proposed circle loss with excellent results achieved for the fine-grained classification tasks. It can be integrated into the proposed ProxylessKD to further boost the performance in the general classification tasks.

## 4 EXPERIMENTS

### 4.1 IMPLEMENTATION DETAILS

**Datasets.** We adopt the high-quality version namely MS1MV2 refined from MS-Celeb-1M dataset (Guo et al., 2016) by Deng et al. (2019a) for training. For testing, we utilize three face verification datasets, i.e. LFW (Huang et al., 2008), CPLFW (Zheng & Deng, 2018), CFP-FP (Sengupta et al., 2016). Besides, we also test our proposed method on large-scale image datasets MegaFace (Kemelmacher-Shlizerman et al., 2016), IJB-B (Whitelam et al., 2017) and IJB-C (Maze et al., 2018)). Details about these datasets are shown in Table 1.

| Datasets | #Identity | #Image |
|----------|-----------|--------|
| MS1MV2 | 85K | 5.8M |
| LFW | 5,749 | 13,233 |
| CPLFW | 5,749 | 11,652 |
| CFP-FP | 500 | 7,000 |
| MegaFace | 530(P) | 1M(G) |
| IJB-B | 1,845 | 76.8K |
| IJB-C | 3,531 | 148.8K |

Table 1: Face recognition datasets for training and testing. (P), (G) mean the probe, gallery set, respectively.

**Data processing.** We follow Wang et al. (2018b); Deng et al. (2019a) to generate the normalized face crops ($112 \times 112$) with five facial points in the data processing. All training faces are horizontally flipped with probability 0.5 for data augmentation.

**Network architecture.** In this parper, we set the n=4, i.e., the four models are the ResNet152, ResNet101, AttentionNet92 and DenseNet201 as the ensemble of teacher models, and choose ResNet18 as the student model. After the last convolutional layer, we leverage the FC-BN structure to get the final 512-D embedding. In the ensemble procedure of four teacher models, we train again a new dimensionality reduction layer and the classifier layer with the feature that is cascaded

from four teacher models as input to acquire a new embedding. This new embedding is applied to knowledge distillation in L2KD, and the new classifier is inherited by student model to do knowledge distillation.

**Training.** All models are trained from scratch with NAG (Nesterov, 1983) and 512 batch size for each teacher training and 1024 for the remaining training procedure. The momentum is set to 0.9 and the weight decay is 4e-5. The dimension of all embedding is 512. The initial learning rate is set to 0.1, 0.1, 0.9, 0.35 in the training of the teacher, student, L2KD, and ProxylessKD, respectively. The training process for the teacher model is finished with 8 epochs, and 16 epochs is used for all remaining experiments. We use the cosine decay in all training and Arcface loss as the supervision, in which m = 0.5 and s=64 following Deng et al. (2019a). All experiments are done on 8x2080Ti GPUs in parallel and implemented by the Mxnet library based on Gluon-Face.

**Testing.** In practical face recognition or image search, the embeddings of database usually are extracted by a larger model while the embeddings of query images are are extracted by a smaller model. Considering this, we should evaluate the consistency of embeddings extracted by the larger model and the smaller model, which represents the performance of different KD methods. Specifically, in the identification task, a large model is used to extract the embeddings of the database, and a small model is used to extract embeddings of the query images. In the verification task, we firstly calculate the verification accuracy using embeddings of image pair extracted by the large and the small model respectively, then calculate the verification accuracy using embeddings of image pair extracted by the small and the large model in turn, finally take the average of them. In particular, we use the same verification method on the small datasets (i.e., LFW, CPLFW, and CFP-FP) following Wang et al. (2018b); Deng et al. (2019a). We use two kinds of measure methods (i.e., verification and identification) to test IJB-B, IJB-C dataset and MegaFace dataset. Note the images in the 1M interference set are extracted by the larger model, and the query images are extracted by the small model in the MegaFace dataset. Meanwhile, we introduce performance when we only use a small model.

## 4.2 ABLATION STUDY

**Results on different loss functions.** Our ProxylessKD can be easily combined with the existing loss functions (e.g., L2softmax (Ranjan et al., 2017), Cosface loss (Wang et al., 2018a), Arcface loss (Deng et al., 2019a)) to supervise the learning of embedding and achieve better embedding space alignment. In Table 2, 3, 4, we show the performance difference of our method under the supervision of different loss functions.

As shown in Table 2, 3, 4, compared with L2softmax and Cosface loss, our method achieves better results with the supervision of the Arcface loss function. This proves that our ProxylessKD is able to lift performance by incorporating a powerful loss function. Hence, we can foresee that our method will achieve better results with the development of more powerful loss functions in classification.

| Method | LFW | CPLFW | CFP-FP |
|---|---|---|---|
| ProxylessKD + L2softmax | 99.50 | 90.45 | 94.55 |
| ProxylessKD + Cosface | **99.66** | 90.46 | 93.83 |
| ProxylessKD + Arcface | **99.66** | **90.55** | **93.95** |

Table 2: The evaluation results of different loss functions on LFW,CPLFW and CFP-FP

| Method | MegaFace | |
|---|---|---|
| | Rank-1 | Ver(%)FAR1e-6 |
| ProxylessKD + L2 softmax | 84.94 | 87.66 |
| ProxylessKD + Cosface | 95.55 | 96.15 |
| ProxylessKD + Arcface | **95.80** | **96.21** |

Table 3: The evaluation results of different loss functions on MegaFace dataset

**Results on different margins.** As shown in Figure 3, to further illustrate the impact of different margins on different scale datasets, we utilize Arcface loss with different margins as the supervised loss of the proposed ProxylessKD to test its sensitivity to margins. Red points mark the best results.

| Method | IJBB | | | IJBC | | |
|---|---|---|---|---|---|---|
| | 1-1 | | 1-N | 1-1 | | 1-N |
| | @FAR=1e-4 | @FAR=1e-6 | Top1 | @FAR=1e-4 | @FAR=1e-6 | Top1 |
| ProxylessKD + L2 softmax | 91.93 | 24.40 | 92.57 | 93.97 | 82.57 | 93.78 |
| ProxylessKD + Cosface | 92.91 | 39.81 | 93.57 | 94.70 | **88.15** | 94.82 |
| ProxylessKD + Arcface | **93.05** | **41.07** | **93.75** | **94.71** | 87.90 | **94.88** |

Table 4: The evaluation results of different loss functions on IJB-B and IJB-C dataset.

Specifically, from Figure 3 (a), (b), and (c), we observe that margin=0.2 achieves better results than others. Though on the LFW dataset the best result is gained at margin=0.4, the gap is tiny and merely 0.02%, which demonstrates the small margin is more appropriate at the small scale dataset. However, on the IJB-B/C and MegaFace that are large scale datasets, we find larger margins bring better results. In particular, when the margin is set to 0.5, the same as the setting in training ensemble of teacher's classifier, the performance is the best. This indicates the performance of ProxylessKD will be better if using a larger margin at a large scale dataset, as shown in Figure 3 (d) ∼ (k).

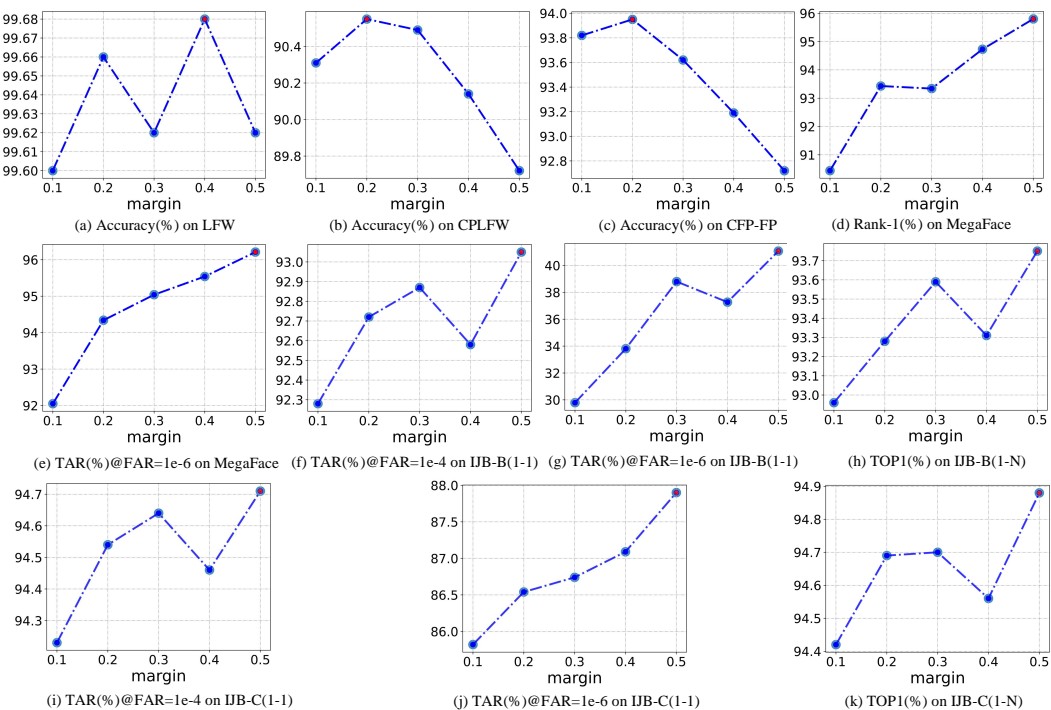

Figure 3: The evaluation of different margins in Arcface loss function

### 4.3 COMPARING WITH OTHER METHODS

**Single and multiple model mode.** We show two evaluation modes in Table 5, 6, 7. L2KD-s and ProxylessKD-s mean the embeddings are only extracted by the student model (single model mode), and the suffix of "-m" represents the embeddings of database are extracted with a teacher model while the embeddings of query images are extracted by the student model (multiple model mode). The more detailed information can be found in Sec. 4.1.

As shown in Table 5, the accuracy on LFW is similar between L2KD and ProxylessKD, but our ProxylessKD achieves better performance on CPLFW and CFP-FP under two evaluation modes. In particular, we achieve 0.91% and 1.32% improvements on CPLFW and CFP-FP under single model mode, and 1.02% and 0.74% better than the L2KD under multiple model mode. Note that ProxylessKD is trained with margin=0.2 with Arcface loss and the margin is set to 0.5 in the next experiments.

| Method | LFW | CPLFW | CFP-FP |
|---|---|---|---|
| Teacher | 99.73 | 93.00 | 97.63 |
| Student | 98.68 | 85.20 | 89.50 |
| L2KD-s | **99.65** | 87.82 | 90.51 |
| ProxylessKD-s | 99.61 | **88.73** | **91.88** |
| L2KD-m | **99.72** | 89.98 | 94.3 |
| ProxylessKD-m | 99.70 | **91.00** | **95.04** |

Table 5: Verification performance (%) of single model mode(-s) and multiple model mode(-m) on LFW, CPLFE, CFP-FP datasets.

| Method | Megaface | |
|---|---|---|
| | Rank-1 | Ver(%)FAR1e-6 |
| Teacher | 97.86 | 97.98 |
| Student | 62.15 | 58.85 |
| L2KD-s | 91.42 | **93.50** |
| ProxylessKD-s | **92.45** | 93.32 |
| L2KD-m | 95.67 | **96.22** |
| ProxylessKD-m | **95.80** | 96.21 |

Table 6: Face identification and verification evaluation results of single model mode(-s) and multiple model mode(-m) on MegaFace. Rank-1 is face identification accuracy with 1M distractors, and "Ver" refers to the face verification TAR at $10^{-6}$ FAR.

The MegaFace dataset contains 100K photos of 530 unique individuals from FaceScrub (Ng & Winkler, 2014) as the probe set and 1M images of 690K different individuals as the gallery set. On MegaFace, we employ two testing tasks (verification and identification) under two mode (i.e., single model mode and multiply model mode). In the testing, except the features of gallery images extracted by the teacher model, the features of the images input the gallery each time is also extracted by the teacher model. In Table 6, we show the performance of L2KD and ProxylessKD. Though MegaFace is a larger scale dataset, and for a more complex recognition task, our proposed method still achieves better results in Rank-1 and boosts the performance by 1.03% and 0.12% under the single model mode and multiple model mode, respectively. And, in multiple model mode evaluation, it performs better than single model mode.

| Method | IJB-B | | | IJB-C | | |
|---|---|---|---|---|---|---|
| | 1-1 | | 1-N | 1-1 | | 1-N |
| | @FAR=1e-4 | @FAR=1e-6 | Top1 | @FAR=1e-4 | @FAR=1e-6 | Top1 |
| Teacher | 93.60 | 46.85 | 94.34 | 95.07 | 54.54 | 95.30 |
| Student | 77.05 | 24.50 | 86.12 | 80.94 | 43.99 | 85.20 |
| L2KD-s | 88.87 | **41.74** | 91.42 | 91.01 | 70.70 | 92.73 |
| ProxylessKD-s | **90.43** | 39.39 | **92.30** | **92.50** | **76.83** | **93.36** |
| L2KD-m | 92.25 | 40.08 | 93.39 | 94.04 | 85.30 | 94.60 |
| ProxylessKD-m | **93.05** | **41.07** | **93.75** | **94.71** | **87.90** | **94.88** |

Table 7: 1:1 verification TAR(%) on (@FAR=1e-4 and @FAR=1e-6) and 1:N identification accuracy-Top1(%).

The IJB-B dataset (Whitelam et al. (2017)) has 1,845 subjects with 21.8K static images and 55 K frames from 7,011 videos. There are 12,115 templates with 10,270 authentic matches and 8 M impostor matches in all. The IJB-C dataset (Maze et al., 2018) is the extension of IJB-B, containing 3,531 subjects with 31.3K static images and 117.5K frames from 11.779 videos. There are 23,124 templates with 19,557 genuine matches and 15,639 impostor matches in all. As shown in Table 7, compared with the L2KD, ProxylessKD achieves the best results among all evaluation methods on IJB-B and IJB-C and

more consistent improvement than L2KD in embedding space alignment between teacher and student. We can see the performance of multiple model mode is better than single model mode, which explains the reason why the base database embeddings are extracted by a large model in the practical

face recognition. Especially at $@FAR = 1e - 6$ on IJB-C, the accuracy of multiple model mode is improved by $10 \sim 15$ % than single model mode. This explains why we advocate directly aligning the embedding space is more essential due to the base database features are also extracted by a large model in the parctical face recoginition. The experiments prove ProxylessKD is a more effective strategy than the existing practical face recognition method (i.e., L2KD).

## 5 CONCLUSIONS

In this work we propose a simple yet powerful knowledge distillation method named ProxylessKD, which inherits the teacher's classifier as the student's classifier to directly optimize the remaining networks of the student while fixing the classifier's weights in the training procedure. Compared with L2KD, which only considers the intra-class distance and ignores the intra-class distance, our proposed ProxylessKD pays attention to them both. Meanwhile, it can benefit from the large margin of existing constraints, which is new to exiting knowledge distillation research. Our method can achieve better performance than other distillation methods in most evaluations, proving its effectiveness.

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
