# OpenReview forum: "ProxylessKD: Direct Knowledge Distillation with Inherited Classifier for Face Recognition"
_ICLR.cc/2021/Conference — Reject_

### Official Review · AnonReviewer4 · 2020-10-28
**Good idea, but need more experiments**

**Rating:** 5
**Confidence:** 3

**Review:**

This paper proposes ProxylessKD method from a novel perspective of knowledge distillation. Instead of minimizing the outputs of teacher and student models, ProxylessKD adopts a shared classifier for two models. The shared classifier yields better aligned embedding space, so the embeddings from teacher and student models are comparable. Since the optimization objective for student model is learning discriminative embeddings,  the face recognition performance is improved compared to the vanilla KL counterpart.

However, I still have some concerns.
First, ProxylessKD makes an assumption that the subjects of the dataset for training both models are somehow overlapping, while L2KD does not have such limitation. So more analysis and detailed discussions on the pros and cons of ProxylessKD and L2KD are needed.

Second, ProxylessKD can be interpreted as initializing classifier of student model by the classifier of teacher model. It would be interesting to see how performance changes with more layers of student model inherited from teacher model. For example, the last two layers, the last three layers.

Third, experimental comparisons with more advanced KD methods are necessary, e.g.[1], [2], [3] etcs. Currently the only comparison with other method is a self-implemented L2KD, which couldn't comprehensively show the effectiveness of the proposed method.
[1] Park, Wonpyo, et al. "Relational knowledge distillation." Proceedings of the IEEE Conference on Computer Vision and Pattern Recognition. 2019.
[2] Peng, Baoyun, et al. "Correlation congruence for knowledge distillation." Proceedings of the IEEE International Conference on Computer Vision. 2019.
[3] Karlekar, Jayashree, et al. "Deep face recognition model compression via knowledge transfer and distillation." arXiv preprint arXiv:1906.00619 (2019).

---

> ### Author Response · Authors · 2020-11-17
> **Thanks for your careful and valuable comments. We will explain your concerns point by point.**
>
> Thanks for your careful and valuable comments. We will explain your concerns point by point.
> **Q1:**
> However, I still have some concerns. First, ProxylessKD makes an assumption that the subjects of the dataset for training both models are somehow overlapping, while L2KD does not have such limitation. So more analysis and detailed discussions on the pros and cons of ProxylessKD and L2KD are needed.
>
> **A1:**
> Compared to the L2KD method, our method indeed requires overlapping datasets. But, ProxylessKD considers not only minimizing the intra-class but also maximizing the inter-class distance. However, the L2KD only considers minimizing the intra-class distance. Meanwhile, our ProxylessKD can benefit from large margin constraints (e.g. Cosface loss and Arcface loss) from which L2KD does not benefit. The experiments also show that our proposed ProxylessKD is more effective than L2KD.
>
> **Q2:**
> Second, ProxylessKD can be interpreted as initializing classifier of student model by the classifier of teacher model. It would be interesting to see how performance changes with more layers of student model inherited from teacher model. For example, the last two layers, the last three layers.
>
> **A2:**
> Because the weight of the teacher's classifier can be approximately viewed as the center of each class and the goal of this paper is to align embedding space between the teacher model and the student model, our ProxylessKD directly inherits the classifier of the teacher model to optimize task. We agree that changing more layers of the student model inherited from the teacher model would be an interesting direction to explore, and we will leave it to future works.
>
> **Q3:**
> Third, experimental comparisons with more advanced KD methods are necessary, e.g.[1], [2], [3] etcs. Currently the only comparison with other method is a self-implemented L2KD, which couldn't comprehensively show the effectiveness of the proposed method.
> [1] Park, Wonpyo, et al. "Relational knowledge distillation." Proceedings of the IEEE Conference on Computer Vision and Pattern Recognition. 2019.
> [2] Peng, Baoyun, et al. "Correlation congruence for knowledge distillation." Proceedings of the IEEE International Conference on Computer Vision. 2019.
> [3] Karlekar, Jayashree, et al. "Deep face recognition model compression via knowledge transfer and distillation." arXiv preprint arXiv:1906.00619 (2019).
>
> **A3:**
> We adopt L2KD as the baseline because it can ensure the student model's embedding space is well aligned with the teacher model's. This is critical for similarity comparison between embeddings across different students model. In the practical application of face recognition, embeddings extracted from different models are required to be in the same embedding space, so that similarity comparison can be conducted. The advanced KD methods only force the student to mimic the teacher’s behavior and do not ensure the embedding space alignment. Thank you for the suggestion and we agree it would be easier for the reader to understand if we add the suggested experiments.

---

### Official Review · AnonReviewer1 · 2020-10-28
**This work has simple idea and is easy to implement. However, novelty is somewhat limited and more comparison shoud be provided.**

**Rating:** 4
**Confidence:** 4

**Review:**

The paper proposes a knowledge distillation method for face recognition, which inherits the teacher’s classifier as the student’s classifier and then optimizes the student model with advanced loss functions.  The paper demonstrates using an ensemble of teacher models can boost the performance of knowledge distillation.

Strength:
- The proposed method is simple and easy to implement.
- The experimental results demonstrate the effectiveness of the technique.
- The paper is well organized and well written.

Weakness:
- The novelty is limited. Directly inheriting the teacher’s classifier is a common strategy in the face recognition community, which can be found in Ref.1.
- The experiment lacks comparison with the general knowledge distillation methods (Ref.2) in image classification and the specific used methods (Ref.3) in face recognition.

【1】Deng, Jiankang, and Guo, Jia et al. Lightweight Face Recognition Challenge. Proceedings of the IEEE/CVF International Conference on Computer Vision (ICCV) Workshops
【2】Hinton, G., Vinyals, O., Dean., J.: Distilling the knowledge in a neural network. In: arXiv preprint arXiv:1503.02531 (2015)
【3】Xiaobo Wang and Tianyu Fu et al. Exclusivity-Consistency Regularized Knowledge Distillation for Face Recognition. ECCV2020

---

> ### Author Response · Authors · 2020-11-18
> **Thanks for your careful and valuable comments. We will explain your concerns point by point.**
>
> Thanks for your careful and valuable comments. We will explain your concerns point by point.
>
> **Q1:**
> The novelty is limited. Directly inheriting the teacher’s classifier is a common strategy in the face recognition community, which can be found in Ref.1.
>
> **A1:**
> Thank you very much for pointing out the related work Ref. 1 that we missed. In Ref. 1, they directly copy and fix the weights of the margin inner-product layer of the teacher model to the student model to train the student model combined with distillation loss. And the motivation of Ref. 1 is the student model can be trained with better pre-defined inter-class information from the teacher model.
>
> Despite the similarity of coping the weight from teacher model, we summaries the following novel contributions that make our work differ from Ref. 1 and existing works:
> 1. We analyze the shortcomings of existing knowledge distillation methods. Specifically, the existing methods target optimizing the proxy task rather than the target task; and they cannot conveniently integrate with advanced large margin constraints to further lift performance. These valuable analyses and observations are not found in Ref. 1 and other existing works.
> 2. Strong motivation and the physical explanation of the proposed ProxylessKD is well explained in our work. Fig 1 and corresponding analysis explained why ProxylessKD can achieve better performance than the existing methods that optimize the proxy task. Such in-depth analysis and strong physical explanation are novel and cannot be found in Ref. 1 and other existing works. We believe these novel findings and the proposed solution are valuable to the face recognition community and will inspire researchers in related fields.
> 3. Solid experiments are designed and conducted to justify the importance of directly optimize the final task rather than the proxy task when doing knowledge distillation. And the properties of ProxylessKD about using different margin-based loss function and hyper-parameters are well examined.(See. Sec 4.2) These detailed analyses about ProxylessKD cannot be found in Ref. 1 and other existing works.
> We believe the above important differences and novel contributions make our work differs from Ref. 1 and existing works. Thank you for point out the missing related work, we have added it to the related work with more detailed discussions.
>
> **Q2:**
> The experiment lacks comparison with the general knowledge distillation methods (Ref.2) in image classification and the specific used methods (Ref.3) in face recognition.
>
> **A2:**
> In face recognition systems, embeddings extracted from different models are required to be mapped into the same embedding space, so that similarity comparison can be conducted across different devices and models in different versions. However, general KD methods only force the student to mimic the teacher’s behavior and thus cannot ensure the embedding space alignment. Ref.2 and Ref.3 belong to the above category and thus cannot meet the requirement for similarity comparison across different models for real-world face recognition systems, and thus not compared.

---

### Official Review · AnonReviewer2 · 2020-10-28
**The paper proposes a novel knowledge distillation (KD) framework of directly using teacher model's classifier to distill the student feature learning. There are several empirical results showing the method's effectiveness.**

**Rating:** 6
**Confidence:** 3

**Review:**

This paper proposes a new KD method to inherit classifier from teacher models and utilize it to train the student model feature representation, where previous KD methods are mostly focusing on the proxy task other than the target task itself.

The idea of using teacher model’s classifier to directly reshape the student model’s feature representation is somewhat novel. It considers the situation of single teacher model and multiple teacher models. The teacher ensemble is achieved by concatenating features from each of the teacher model and then conducting dimension reduction using PCA. The methodology illustration is simple yet clear. There are multiple experiments on major face recognition datasets and demonstrate superior performance against baselines such as L2KD-s.

Regarding the concerns, I am listing them into bullets.

1. why the experiments make the setting of templates using teacher model to extract feature, while the query using student model to extract feature?

Would the comparison of using student model for extracting both template and query feature be possible? It can provide a direct comparison to other methods, i.e. ArcFace trained using ResNet18 compared to the student model with ResNet18.

Current experiments lack the comparison to the state-of-the-art methods, i.e. ArcFace and CosFace.

2. the ablation is emphasizing on the ProxylessKD combining with different losses. It does not consider the knowledge distillation itself. For example, when ProxylessKD is combined with the proxy task, i.e., feature distillation loss, how would it perform compared to only ProxylessKD? Meanwhile, in many KD papers, there are also intermediate layer feature distillation, would it harm the overall performance under this paper's setting? It needs sufficient analysis to justify the authors' choice of only applying the teacher model's classifier as distillation.

3. In ablation, how would the number of teachers influence the student performance? Meanwhile, how would the network architecture influence the student performance? i.e., fixing the teachers to be the same, while varying student architecture with multiple hypothesis, i.e., ResNet, AttentionNet, DenseNet? It is good to know what specific architecture is favored under the authors' proposed framework.

---

> ### Author Response · Authors · 2020-11-17
> **Thanks for your careful and valuable comments. We will explain your concerns point by point.**
>
> Thanks for your careful and valuable comments. We will explain your concerns point by point.
>
> **Q1:**
> why the experiments make the setting of templates using teacher model to extract feature, while the query using student model to extract feature?
>
> **A1:**
> We've designed two experiment settings:
> 1. single model mode: both using student model to extract feature
> 2. multiple model mode: templates using teacher model to extract feature, while the query using student model to extract feature
>
> The first setting is a common setting for evaluating the face recognition of the student model; The second setting aims to check if the student's embedding space is perfectly aligned with the teacher's embedding space, which is very important for making querying across templates extracted between different models possible.
>
> Note that: querying across templates extracted between different models is very important in the practical application of face recognition, for two reasons: 1) ensure feature consistency between model updates; 2) the embeddings extracted from different devices (e.g. smart security camera, mobile phone, CPU Server, GPU Server) with different models can be directly compared.
>
> **Q2:**
> Would the comparison of using student model for extracting both template and query feature be possible?
>
> **A2:**
> It’s sure. We also use the student model to extract both template and query features to illustrate the performance in Table 5, 6, 7 of the paper. E.g. “ProxylessKD-s”. However, we mainly focus on the setting that the template features are extracted by teacher model and the query features are extracted by student model, which is more meaningful for practical application.
>
> **Q3:**
> It can provide a direct comparison to other methods, i.e. ArcFace trained using ResNet18 compared to the student model with ResNet18.
>
> **A3:**
> In Table 5,6,7, “Student” is the “ArcFace trained using ResNet18” and “ProxylessKD-s” can be viewed as “the student model with ResNet18”.  The experiments also show that our proposed ProxylessKD is an effective distillation method.
>
> **Q4:**
> Current experiments lack the comparison to the state-of-the-art methods, i.e. ArcFace and CosFace.
>
> **A4:**
> In Table 5,6,7, “Student” is the “ArcFace trained using the same backbone, which is significantly lower than our proposed method. Besides, models trained directly using Arcface (or CosFace) cannot ensure the embedding spaces are well aligned, and thus querying across templates extracted from different devices and models is impossible.
>
> **Q5:**
> the ablation is emphasizing on the ProxylessKD combining with different losses. It does not consider the knowledge distillation itself. For example, when ProxylessKD is combined with the proxy task, i.e., feature distillation loss, how would it perform compared to only ProxylessKD?
>
> **A5:**
> Multi-task learning (i.e. combine proxy task and proxyless task) is an interesting topic to explore but not the core of our work. We agree that it would interesting to see if L2KD will give a further performance boost for ProxylessKD. Thanks for the suggestion, we are running this experiment and will add it.
>
> **Q6:**
> Meanwhile, in many KD papers, there are also intermediate layer feature distillation, would it harm the overall performance under this paper's setting? It needs sufficient analysis to justify the authors' choice of only applying the teacher model's classifier as distillation.
>
> **A6:**
> The main contribution of this work is to point out the drawbacks of optimizing the proxy task rather than the target task when transferring knowledge from teacher models and present an inspiring method that can directly optimize the target task. We justify our claims under the most commonly adopted experiment settings and focusing on the face recognition task. We do not consider intermediate layer feature distillation, because: 1) it is rarely used for face recognition; 2) we couldn't find a strong baseline to follow. 3) embedding space cannot be well aligned for querying across different models
>
> **Q7:**
> In ablation, how would the number of teachers influence the student performance?
> Meanwhile, how would the network architecture influence the student performance? i.e., fixing the teachers to be the same, while varying student architecture with multiple hypothesis, i.e., ResNet, AttentionNet, DenseNet? It is good to know what specific architecture is favored under the authors' proposed framework.
>
> **A7:**
> Our work does not focus on multi-model ensembling and thus does not have ablation studies to evaluate how would the number of teachers influence student performance. We would suggest the readers to refer multi-model ensembling related works for more information.
> The design of loss function is usually orthogonal student architectures based on previous works. Since our proposed method is not designed for specific CNN architecture, we adopt the most commonly used ResNet without loss of generality. Thanks for the suggestion.

---

### Decision · Program_Chairs · 2021-01-07
**Final Decision**

**Decision:**

Reject

**Comment:**

This paper presents a knowledge distillation method for face recognition, by inheriting the teacher’s classifier as the student’s classifier and optimizing the student model with advanced loss functions. It received comments from three reviewers: 1 rated “Ok but not good enough - rejection”, 1 rated “Marginally below” and 1 rated “Marginally above”. The reviewers appreciate the simple yet clear methodology illustration and the well written paper. However, a number of major concerns are raised by the reviewers, including limited novelty, lack of comparison with more advanced knowledge distillation methods and their special case in face recognition. During the rebuttal, the authors made efforts to response to all reviewers’ comments. However, the rating were not changed. The ACs concur these major concerns and more comprehensive comparisons with the state of the art KD methods are necessary to better illustrate the contribution of this work. Therefore, this paper can not be accepted at its current state.